# The evolution of human population distance to water in the USA from 1790 to 2010

Yu Fang [1,2] & James W. Jawitz [1]

Human societies evolved alongside rivers, but how has the relationship between human settlement locations and water resources evolved over time? We conducted a dynamic analysis in the conterminous US to assess the coevolution of humans and water resources from 1790 to 2010. Here we show that humans moved closer to major rivers in pre-industrial periods but have moved farther from major rivers after 1870, demonstrating the dynamics of human reliance on rivers for trade and transport. We show that humans were preferentially attracted to areas overlying major aquifers since industrialization due to the emergent accessibility of groundwater in the 20th century. Regional heterogeneity resulted in diverse trajectories of settlement proximity to major rivers, with the attractiveness of rivers increasing in arid regions and decreasing in humid areas. Our results reveal a historical coevolution of human-water systems, which could inform water management and contribute to societal adaptation to future climate change.

[1] Soil and Water Sciences Department, University of Florida, Gainesville, FL 32611, USA. [2] State Key Laboratory of Hydro-science and Engineering, Department of Hydraulic Engineering, Tsinghua University, 100084 Beijing, China. Correspondence and requests for materials should be addressed to J.W.J. (email: jawitz@ufl.edu)

Water is an attractive factor when people choose where to live. Historically, humans have chosen to live close to rivers for domestic and agricultural water supply as well as for navigation purposes, which has led humans to follow the courses of rivers during migrations[1–3] and locate in proximity to rivers when establishing settlements[4,5]. However, humans have been able to decrease their reliance on direct proximity to rivers by developing advanced measures to transport adequate water from other sources (for example, canals or pipelines, groundwater pumping, and desalination)[6–9], and shifting from waterborne transport modes to land (railways and roads) and air transport[10]. Meanwhile, humans are more vulnerable to flood risk when settling adjacent to rivers and thus require development of flood protection measures to reduce flooding risks[11]. The importance of locating close to rivers has been increasingly reflected by the indirect environmental amenity value of water and the agglomeration economic effects of historical established settlements in proximity to rivers[12–14]. Furthermore, groundwater has gradually emerged as a critical water source, largely due to its convenience, availability, and quality[15,16]. Since the 1940s and 1950s, groundwater extraction in the US has increased remarkably as groundwater pumping became economical and efficient due to the development of technology and widespread access to electricity[15,17].

The dynamic and complex human–water interactions embedded within heterogeneous landscapes present significant questions about where humans live in relation to water resources. The relationship between human settlement density and distance to rivers has been analyzed at high spatial resolution at a global scale[5,18], finding denser human population with increased proximity to rivers, but with regional variability based on climate, degree of urbanization, and economic development history. However, these studies were conducted as a temporal snapshot (for 2007)[5], or with a relatively narrow temporal window (between 1992 and 2013)[18]. Also, the spatial units of analysis at continental or country scale[18] may not provide sufficient perspective for smaller areas, such as catchments. Therefore, the changing role of water resources in regulating where people live over longer timescales (e.g., centuries) remains to be examined, and an integrated approach in space and time is required to answer this question quantitatively.

A significant obstacle is the availability of spatially explicit population data over long time periods. Spatially explicit population maps, including the Gridded Population of the World (GPW)[19], the Global Rural Urban Mapping Project (GRUMP)[20], LandScan[21], and WorldPOP[22], are of limited utility for long-term (e.g., multi-decadal) dynamic analyses since they only became available after the 1990s, while historical census data for the conterminous US, although available from 1790 to 2010, lack detailed information within counties[23]. However, a recent reconstruction of historical human population distribution in the conterminous US[24] based on decennial census data has provided an opportunity to conduct a quantitative spatio-temporal analysis of the coevolution of human settlement locations and water.

Here, we studied the spatio-temporal relationship between human populations and water resources in the conterminous US for the decades 1790–2010 (excluding 1960, for which population data are lacking)[24]. During this time, large migrations of primarily European settlers gradually expanded from the Atlantic coast to the Pacific coast[3], and the United States transitioned from a pre-industrial, to industrial, then to post-industrial society. The Second Industrial Revolution, generally recognized as the period between 1870 and 1914, with some characteristic events dated to the 1850s, led to large expansions of railroad lines, increased use of machinery and steam power, and the onset of electrification[25]. We asked two major questions about how societal relationships with water resources (rivers and aquifers) evolved in correspondence with these technological transformations. First, are humans getting closer to or farther from major rivers? Second, how is the relationship between human settlement locations and groundwater availability changing over time? Additionally, we examined how hydroclimatic spatial heterogeneity has contributed to regionally distinct temporal trends of human distance to water.

We overlaid the reconstructed historical human population data[24] with maps of both major rivers (Fig. 1a) and groundwater accessibility (Fig. 1b) (see Methods) in order to examine the temporal trends of human population distribution across different distance classes to major rivers and among distinct types of aquifers. We used the relationship between normalized population density (NPD) and distance to major rivers (DMR) to indicate the desirability of living in proximity to rivers, and applied the mean population density overlying corresponding aquifers to represent the attractiveness of each aquifer type. The USGS two-digit Hydrologic Unit Code (HUC, Fig. 1c) region boundary shapefile from the Watershed Boundary Dataset (http://nhd.usgs.gov/wbd.html) was adopted for analyses on regional heterogeneity. Additionally, we included geomorphological alteration of river courses over time by incorporating both river width data and the temporal evolution of reservoirs (Fig. 1d) (see Methods for details).

We illustrated our hypothesized dynamic geographic relationships between humans and hydrological systems (major rivers and aquifers) in Fig. 2. First, we hypothesize that during the pre-industrial period, as frontier human settlements grew to small cities, the desirability of locating adjacent to major rivers increased due to the strong reliance on waterborne transport and naturally available water resources provided by nearby rivers (Fig. 2a). Second, we hypothesize that following industrialization, the relative importance of major rivers for influencing human settlement locations has generally abated (Fig. 2b), while electrified pumping has made groundwater more accessible, resulting in greater population growth in regions overlying productive aquifers (Fig. 2c). Our results support both hypotheses. From 1790 to 1870, for the entire conterminous US, humans moved steadily closer to major rivers. Starting in 1870, and continuing through the period of record (until 2010), this trend reversed with humans moving progressively farther away from major rivers. The expansion of land-based transport networks from the late 1800s together with growing access to groundwater throughout the 20th century effectively unshackled humans from their historic requirements of proximity to major rivers.

## Results

**Importance of major rivers**. In the conterminous US, the calculated reference DMR (Fig. 1a) varied across regions, with a larger portion of land found within small DMR in the east humid region, due to the relatively denser river system, compared to the western arid region. To account for such natural hydrographic differences, the ratio of human distance ($D_H$) to geographical distance ($D_G$) was used to evaluate the relative human DMR among regions. Values of $D_H/D_G < 1$ indicate a human preference for settling closer to rivers than would be suggested by the land distance between rivers[5]. Based on the temporal snapshot analysis for 2010 for the 19 HUC regions of the conterminous US, $D_H/D_G = 0.62 \pm 0.24$, with $D_H/D_G \leq 1$ for all HUCs, indicating an overall pattern of human preference for proximity to rivers in 2010. The range in $D_H/D_G$ found here, from 0.20 in HUC 14 to 1.00 in HUCs 3, 8, and 12 (Supplementary Fig. 1a and 1b), is different from the range of 0.8–1.3 reported for a snapshot from 2007 for 12 food producing units in the US[5], possibly because of the

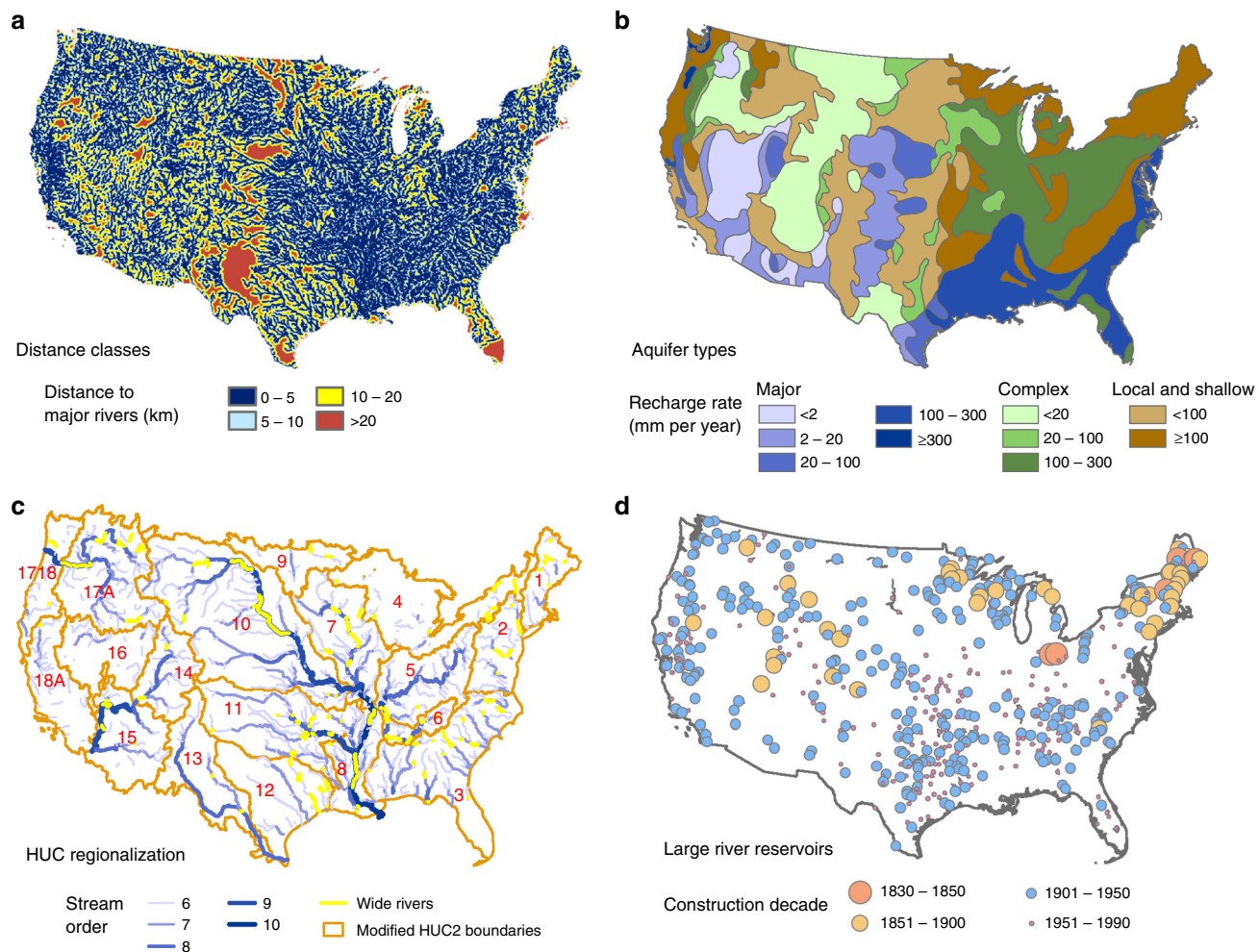

**Fig. 1** Rivers, aquifer types, reservoirs, and regions in the conterminous US. **a** Calculated distance to major rivers (stream order ≥ 4) for each pixel (1 km × 1 km) considering river width. **b** Aquifer types based on geologic structure and groundwater recharge rate. **c** Selected major rivers (stream order ≥ 6) and the modified boundaries for Hydrologic Unit regions. Yellow indicates rivers wider than 1 km according to the GWRL database. **d** Large river reservoirs over time

differences in resolution of human population and stream network datasets between studies. Additionally, $D_H/D_G$ was negatively related with mean annual precipitation ($R^2 = 0.30$, $p < 0.01$, Supplementary Fig. 1a), indicating that the preference for settling near major rivers was greater in arid regions.

To examine dynamic changes in human settlement locations with respect to rivers, we computed the relationship between population density and distance to rivers at decadal intervals from 1790 to 2010. For the entire conterminous US, NPD decreased as DMR increased, and this relationship was consistently negative over time. However, the absolute value of the slope increased from 1790 to 1870 (Fig. 3a) and then decreased after 1870 (Fig. 3b). The turning point, 1870, is consistent with the beginning of the Second Industrial Revolution[25]. These dynamics of the slope of NPD vs. DMR (Fig. 3c), demonstrate that major rivers became more important for human settlement locations prior to 1870, with decreasing importance afterwards.

**Importance of groundwater**. Higher population densities were found above aquifers with higher recharge rates (Fig. 3d). Major aquifers with high recharge rates (>100 mm per year, aquifer types 14 and 15), were the most attractive for humans in the 20th century. The major aquifers with the highest recharge rate

(aquifer type 15) supported the highest population densities and experienced the fastest growth in population density (Fig. 3d and Supplementary Table 1). The differences in population density between aquifer type 15 and the other nine types of aquifers increased steadily, but with a turning point identified in the 1940s ($p < 0.05$), when groundwater pumping became more efficient and advanced[15]. Generally, mean population density increased with potential water supply from underlying aquifers. With similar recharge rates, population density tended to be higher over major aquifers, and lower over local and shallow aquifers.

Groundwater supplemented surface water and thus alleviated the desirability of living close to major rivers. Across the US, areas with relatively low-yielding local and shallow aquifers (types 33 and 34) were associated with strong desirability of living close to rivers (NPD vs. DMR slope ≤ −0.03, Supplementary Fig. 2 and Supplementary Table 2), regardless of recharge rates. Conversely, complex and major aquifers with high recharge rates (aquifer types 24, 13, 14, and 15) had relatively lower desirability of living close to rivers (slope ≥ −0.02). Major aquifers with recharge rates of 100–300 mm per year (aquifer type 14) had the lowest desirability of living close to major rivers, with NPD positively related with DMR, indicating the role of high-recharge major aquifers in facilitating the decoupling of the historical proximity of human settlements to rivers.

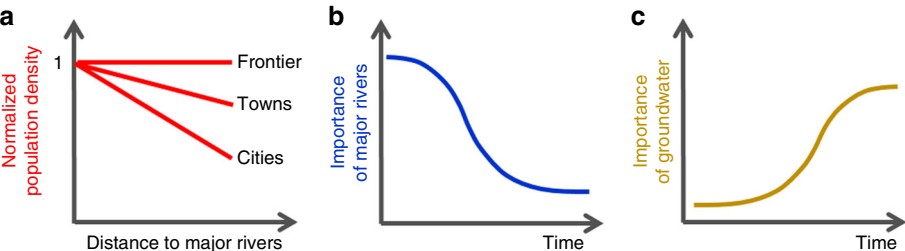

**Fig. 2** The hypotheses tested in this study. **a** Hypothesized relationship between normalized population density (NPD) and distance to major rivers (DMR), and their temporal trends associated with the development of settlements from frontier, to towns, and to cities, in pre-industrial societies. The dynamics of the importance of **b** major rivers and **c** groundwater for human settlement locations after industrialization

**Regional heterogeneity**. Human distance to rivers exhibited regional heterogeneity at all temporal intervals examined, with distinct regional temporal trends (Supplementary Fig. 3). We identified three major typologies of temporal changes in NPD vs. DMR, and illustrated each using a typical HUC region (Fig. 4). The first type, with temporally constant desirability of settlement proximal to major rivers, was found for HUCs 1, 2, 11, 12 and is illustrated for HUC 1 from 1850 to 2010 in Fig. 4a, where the slopes between NPD and DMR were approximately steady (Fig. 4d). This relative stability may be indicative of an equilibrium state that follows the evolution from frontier to cities illustrated in Fig. 2a. For example, by 1850 the New England settlements of HUC 1 had already been long-established with relatively little structural change since then.

The second type, with the absolute value of negative slopes decreasing as the slopes approach zero, and even increasing to become positive slopes, is exemplified by HUC 3 (Fig. 4b, e) (also found in HUCs 7, 8, and 17, 18, Supplementary Fig. 3). This indicates decreasing desirability of proximity to rivers, with humans moving farther from rivers, which could be associated with relatively later settlement histories and thus continued evolution from towns to cities (Fig. 2a) during the study period. However, this pattern could also be influenced by the presence of abundant groundwater resources (Fig. 2c) as in HUC 3.

The third type, with slopes between NPD and DMR becoming increasingly steeper, indicating increased attractiveness of living close to major rivers, is illustrated by HUC 15 (Fig. 4c, f) (also found in HUCs 4, 6, 9, 13, 14, 16, 17A, Supplementary Fig. 3). This pattern is mostly associated with arid regions and highlights that under conditions of water scarcity, the general hypotheses described in Fig. 2 appear to be outweighed by the continued attractiveness of proximity to rivers. In addition to these major types, four HUCs showed combinations of these trends, with desirability of proximity to rivers in HUC 5 first increasing then decreasing, while HUC 10 and 18A regions showed the opposite since 1850.

## Discussion

Societal progress and technological development have greatly changed how people obtain water resources[7]. Are humans also changing where they live in relation to water? This study provided a quantitative analysis of human water coevolution over a timescale of centuries. For the conterminous US, our analysis demonstrated a persistent preference of living close to major rivers, with consistently negative relationships between population density and DMR from 1790 to 2010 (Fig. 3a, Fig. 3b). We also quantified the dynamic desirability of living adjacent to rivers using the linear slopes of NPD and DMR, which echoed changing societal reliance on adjacency to major rivers over time.

In pre-industrial society, humans were highly reliant on proximity to rivers for agriculture, water supply, and navigation[5]. With industrialization, human settlements transformed their transport modes from primarily water-borne to other means such as roads and railroads[10]. As human settlements have continued to grow, water demand in some cities has increased beyond the local supplies, as illustrated by cases such as Los Angeles, Phoenix, and Atlanta[7]. Thus, water supply in post-industrial societies has evolved from people moving to the water to people moving the water[26]. For example, in Los Angeles since 1900, as the population has grown, the water supply from the local Los Angeles River became limiting and the city began to import water from hundreds of kilometers away, including the Owens Valley in 1913, Mono Basin in 1940, the Colorado River in 1941, and the rivers of the Central Valley in 1971[7]. Modern cities can bring adequate water from previously remote and inaccessible sources by building canals or by groundwater pumping[8,9,27,28], increasing water availability and greatly reducing the portion of the total population considered at risk for water scarcity[26]. Around 80% of large cities globally would have to travel less than 22 km to reach a potential unstressed water source adequate for several million people[9]. Thus, water transfers have contributed to the increase in the distance of cities from water sources. Consequently, the desirability of living close to rivers gradually declined and people are no longer constrained to live only in water-rich places, with burgeoning cities also existing in arid regions[29,30]. Thus, the increasing human distance to water in recent decades reflects a process of telecoupling[31] in human–water interaction.

We identified three types of human–water coevolution (Fig. 4 and Supplementary Fig. 3) that were associated with regional differences in climate, water resource endowments, and settlement history. Regions (here based on HUCs) such as New England, with long settlement history and early industrialization, had stable distances from rivers with early evolution from frontier to cities (Fig. 2a). Regions with temporally increasing desirability of proximity to rivers were located in the arid west, while regions with decreasing desirability were mostly found in humid regions with less vulnerability to water scarcity. Additionally, the correlation between fast growth in population density and high-recharge major aquifers suggested an increasing attractiveness of places overlying productive aquifers. We found consistently high mean population density in areas overlying rapidly recharged major aquifers, even after controlling for mean annual temperature (Supplementary Table 3).

Our analysis was conducted at a continental scale over a time period of hundreds of years and is thus subject to the limitation of currently available data. Our analysis did not evaluate water quantity (for example by differentiating by stream order) or quality, which could affect the attractiveness to human settlements. We compared published river course migration rates, and

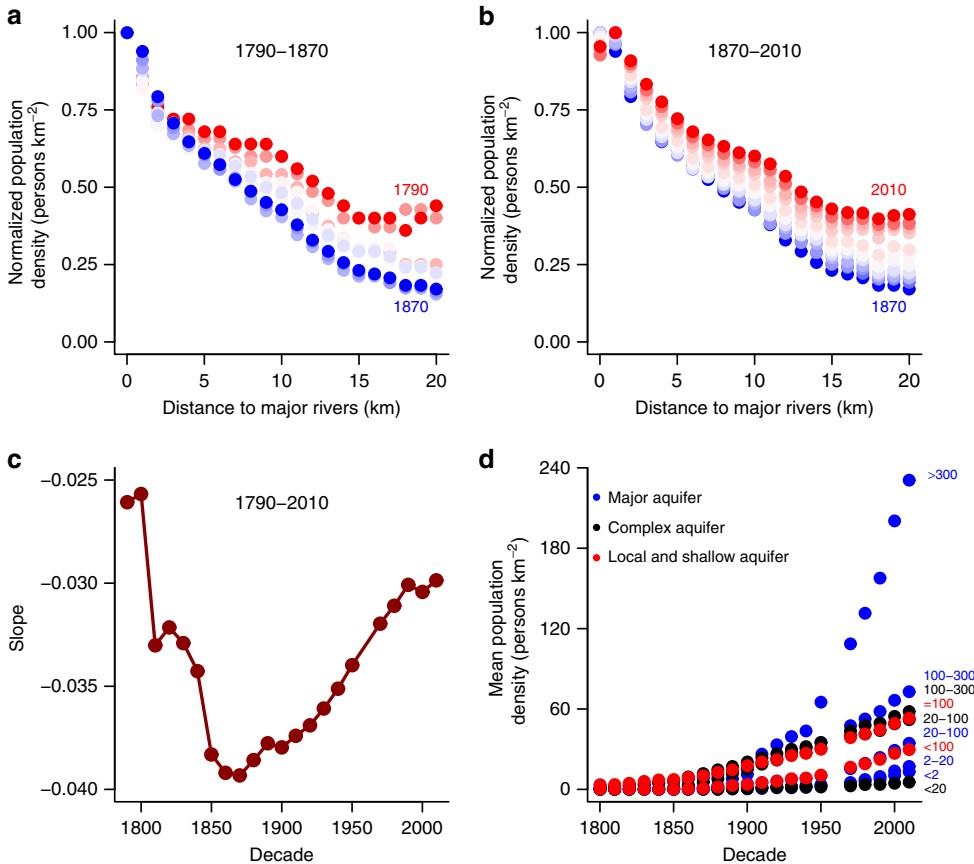

**Fig. 3** The dynamic human population distance to water. The relationship between normalized population density (NPD) and distance to major rivers (DMR) in the conterminous US from **a** 1790 to 1870 and **b** 1870 to 2010. **c** The changing desirability of living close to major rivers, reflected in the slope of NPD vs. DMR. **d** Mean population density overlying each aquifer type from 1790 to 2010. Recharge rates, mm per year, are shown offset. We use the red-to-blue gradient to indicate the trend for the decades to 1870 in **a** and the gradient from blue to red for decades from 1870 in **b**. The data points for **a**, **b** are based on DMR classes while those for **c** and **d** are based on census year

found overall migration rates to be much smaller than the 1-km resolution of our analyses (see Supplementary Note and Supplementary Tables 4 and 5 for details). Continental-scale information about aquifer structure and recharge rates have only relatively recently become available, however, these data are still at much lower spatial resolution than corresponding information about river networks[32]. The timescales for large-scale changes in whole-river network structures and aquifer boundary migration are geological rather than human in scale[33,34]. Thus, at this spatio-temporal scale the assumption of stationarity of river networks and groundwater seems reasonable. However, the importance of aquifer volume changes due to groundwater pumping has been growing in recent decades, and is expected to continue to grow in significance in coming decades[35]. The potential changes of aquifer boundaries and storage volumes due to groundwater withdrawal should be given consideration when projecting human distance to water in the future.

Finally, we note that the dynamic human distance to water also implies changing anthropogenic impacts on water resources over time. Human settlements located adjacent to rivers exert direct pressure on local water quality, resulting in serious impacts to health and ecosystems[36]. Water redistribution[9] has relieved the previous strong dependence on adjacent rivers, but with a high dependence on technology and infrastructure. Meanwhile, the rapid development of some aquifers has contributed to serious groundwater depletion issues, e.g., in the California central valley

and High Plains aquifers[16,27,37]. The shifted human distance to water resources has thus been mainly achieved by co-evolution of infrastructure and water management institutions[38,39]. For example, a majority of US cities have achieved artificial- or super-abundance of water supply through substantial complexity of hydraulic infrastructure institutions[38]. Ensuring adequate infrastructure investment is therefore extraordinarily essential to sustain stable water supplies. However, further infrastructure development to extract water from farther away is becoming more expensive or even infeasible due to the physical, economic, and ecological limits[30,39–41]. Infrastructure projects such as large dams or water transfer projects have also often resulted in forced relocation of people[18,42]. Thus, as water resources become increasingly scarce in the future, whether people move closer to water, or water is moved to the people, feedbacks between these should be considered in future evaluations of the dynamic evolution of the human distance to water.

## Methods

**Data**. For human settlement data, we used the historical human population distribution previously reconstructed for the conterminous US[24] based on the decennial census data, which has a spatial resolution of 1 km and a temporal span from 1790 to 2010 (except 1960) and thus provides a possibility to conduct a dynamic analysis in space and time. Note our analysis excluded 1960 because these data are still not available due to the unusually restrictive data suppression strategy employed by the 1960 Census (https://www.nhgis.org/user-resources/faq#1960_Data). This spatio-temporally explicit population data (model M5 in

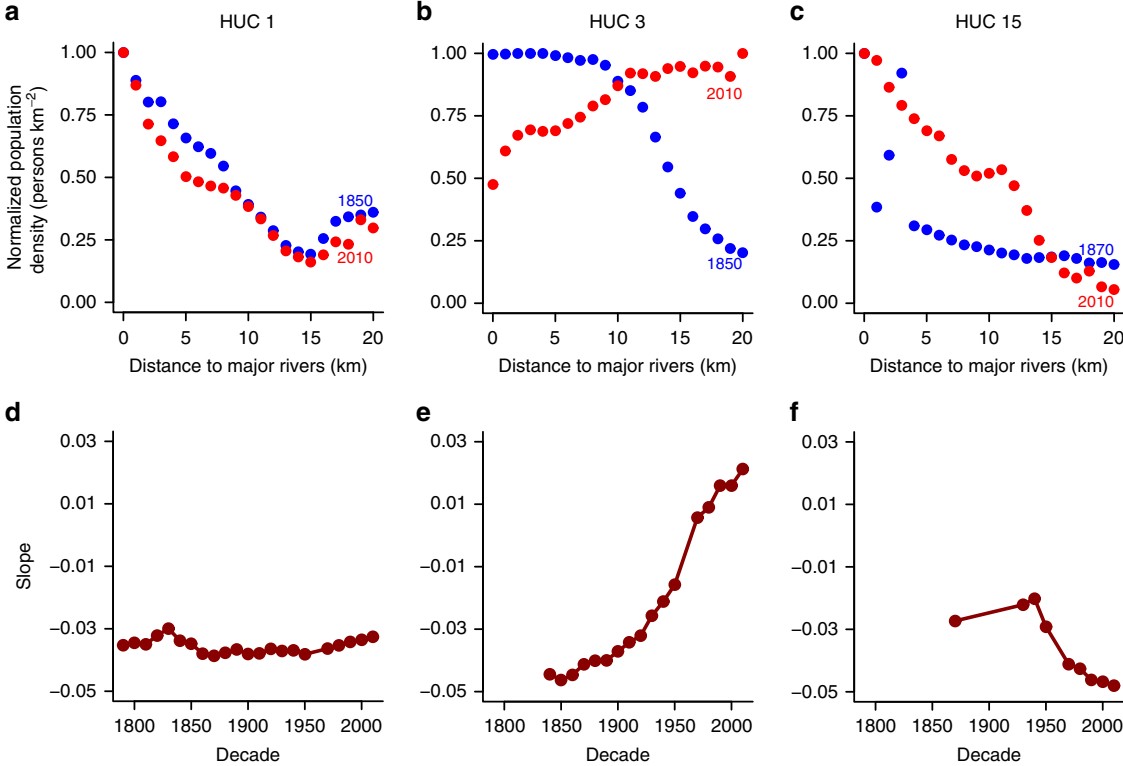

**Fig. 4** Three major types of trajectories of human water coevolution. **a** HUC 1, 1850–2010, **b** HUC 3, 1850–2010, and **c** HUC 15, 1870–2010. **d–f** The dynamics of the slope of normalized population density (NPD) vs. distance to major rivers (DMR) for HUC 1, 3, and 15, 1790–2010. Only slopes with $p < 0.05$ during the settlement periods are shown. The data points for **a–c** are based on DMR classes while those for **d–f** are based on census year

ref. [24]) was generated by separating urban and rural areas and considering multiple factors including non-inhabitable regions, topological suitability, and economic desirability. Neither river networks nor groundwater was involved in population mapping, which allows an un-biased examination of human-water coevolution over time.

For hydrographical data, we used river networks data from 1:100 K National Hydrography Database (NHD) Plus dataset (http://www.horizon-systems.com/nhdplus/, Fig. 1a), USGS two-digit HUC (Fig. 1c) region boundary shapefile from Watershed Boundary Dataset (http://nhd.usgs.gov/wbd.html), and groundwater data from Groundwater Resources Map of North America developed in UNESCO's World-wide Hydrogeological Mapping and Assessment Program (WHYMAP, http://www.whymap.org)[32]. Groundwater resources were divided by WHYMAP into three categories according to geologic structure: major groundwater basins, aquifers with complex hydrogeological structures, and areas with local and shallow aquifers. Among these, major aquifers provide the best conditions for groundwater exploitation. Each type of aquifer was further divided into sub-categories based on recharge rate (Fig. 1b). We used river width data from Global River Widths from Landsat (GRWL) Database[43] to characterize the coverage of rivers and adopted Global Reservoir and Dam (GRanD) database from Socioeconomic Data and Applications Center (http://sedac.ciesin.columbia.edu/data/set/grand-v1-dams-rev01) to reflect the dynamic river coverage and update the distance to rivers over time. We also used the waterway data from U.S. Army Corps of Engineers Navigable Waterway Network (http://www.rita.dot.gov/bts/sites/rita.dot.gov.bts/files/publications/national_transportation_atlas_database/2014/polyline) to help determine the threshold of stream order for major rivers, and applied a climate zone map based on the IPCC classification derived from European Soil Data Center (ESDAC, http://esdac.jrc.ec.europa.eu/projects/renewable-energy-directive) to test the effect of groundwater in influencing human settlements.

**Major river definition and DMR calculation.** Rivers of different orders hold varied roles in human society. We defined rivers with stream order ≥ 4 as major rivers, considering both the discharge required to satisfy water demand for settlements to develop into cities[26] and navigation requirement (fourth order is the minimum in the Navigable Waterway Network). Due to the varied resolutions of NHD and GRWL river shapefiles, we used GRWL data to identify the rivers with mean width larger than 1 km (a total of 7524 km of rivers) and assigned their river width values to the NHD rivers within the width buffer zones of the GRWL wide rivers (Fig. 1c). Minor manual modification was necessary to determine the

coverage of major rivers. We converted the coverage shapefile of major rivers to raster and calculated the Euclidean distance to the closest major rivers (DMR) for each grid cell in the conterminous US at a resolution of 1 km, which was used as the reference DMR under natural condition (Fig. 1a). The threshold of the maximum distance was set as 20 km, since about 95% of the total study area was within this limit (Supplementary Fig. 4). We discretized the distance into 21 classes using 1 km as the interval (DMR = 0, 1, 2, …, 20 km), with the distance of each class representing its maximum DMR.

We considered the changes of river courses over time from the construction of reservoirs (Fig. 1d, Supplementary Fig. 5) using the reservoir location and decade of construction from the GRanD database. This resulted in a maximum increase of about 2.6% in the areas with DMR of zero, compared to the reference DMR based on GRWL data.

**Importance of major rivers and groundwater.** For each DMR class, we summarized the total area ($A$) and total population ($P$), and identified the distances corresponding to 50% of accumulative area and 50% of accumulative population (as illustrated for HUCs 3 and 14 in Supplementary Fig. 1b) and denoted them as geographical distance ($D_G$) and human distance ($D_H$), respectively. The ratio ($D_H/D_G$) was used in a previous study[5] and applied here to compare the relative DMR among regions. Additionally, we calculated the NPD by dividing population density ($P/A$) by its maximum value in the study region. Statistically significant ($p < 0.05$) linear regression slopes between NPD and DMR were used to indicate the desirability of living in proximity to major rivers.

Similarly, we summarized the total area ($A_{GW}$) and total population ($P_{GW}$) overlying with different categories of aquifers, and employed the associated mean population density (MPD = $P_{GW}/A_{GW}$) to represent the attractiveness of each aquifer type. We fit the time series of mean population density with a power function (MPD = $r_0 r^{t-t_0}$, for $t_0 \leq t \leq 2010$) and calculated the growth rate ($r$) for each aquifer type. The initial year, $t_0$, was based on the settlement history of each aquifer type. We used the Pettitt test to identify the turning point of the dynamic differences in mean population density between major aquifers with recharge rate >300 mm per year (aquifer type 15) and the remaining nine types of aquifers. We also examined the control of climate by comparing population density overlying distinct aquifer types for the whole conterminous US to that for different climate zones (tropical, warm temperate, and cool temperate zones, from ESDAC). In addition, considering the complex interaction between rivers and aquifers, we

calculated the NPD values associated with DMR for each of the 10 aquifer types and analyzed their combined effect on population density using 2010 as a snapshot.

**Analysis of spatio-temporal characteristics.** We used HUC regions in the conterminous US to analyze spatial variability in human distance to water. Due to the high variability of precipitation in western coastal regions (Supplementary Fig. 6), HUCs 17 and 18 were further divided into two arid regions, designated as HUCs 17A and 18A, and one humid region, HUC 17-18 (Fig. 1c), according to differences in mean annual precipitation. We applied the approach above to explore the dynamic trends of human–water relationships for each HUC from 1790 to 2010 and classified HUCs into different categories based on the trend of their slopes between NPD and DMR. Only statistically significant slopes ($p < 0.05$) between NPD and DMR were used. Due to the westward migration of primarily European settlers, the settled portions of each HUC expanded gradually over time. The initial decade of analysis was determined for each HUC based on the decade when the settled area reached 90% of the total area (Supplementary Table 6 and Supplementary Fig. 7).

**Code availability.** The ArcGIS Model Builder and R code supporting this analysis are available upon request.

## Data availability

The reconstructed historical human population data are available for download at https://doi.org/10.6084/m9.figshare.c.3890191.v1, as described in ref. [24]. All other data, including river networks, river widths, regional boundaries, reservoir locations, and groundwater data are publicly available as described in the Methods. Source Data are provided for Fig. 3, Fig. 4 and Supplementary Figs. 1-6 as a Source Data file.

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

## Acknowledgements

This research was supported in part by USDA National Institute of Food and Agriculture Hatch project FLA-SWS-005461. Y.F. acknowledges support from National Natural Science Foundation of China (Grant No. 71461010701). We are grateful for the suggestions from Drs. Matthew J. Cohen, Xudong Fu, Yang Hong, Kyungrock Paik, P. Suresh C. Rao, and Andrew D. Wickert.

## Author contributions

Y.F. and J.W.J. conceived the ideas. Y.F. conducted the analyses. Y.F. and J.W.J. wrote the paper.

## Additional information

**Competing interests:** The authors declare no competing interests.

