## [Peer Review File · Nature Communications]

Reviewers' comments:

Reviewer #2 (Remarks to the Author):

This article tackles an interesting question, that of the evolving relationship between human populations and water, by combining information on past censuses and on freshwater source location. The methods are well explained and the paper is easy to read.

Regrettably, the authors "assumed temporal stationarity of the spatial distribution of both major rivers and groundwater over time" (manuscript lines 265-266). This is simply not something that happens, which invalidates all their subsequent work: the quantitative aspects are not believable, nor is an estimate of the error given.

The authors should at least reference some geomorphological studies to support their course of action. However, I believe it will be hard to do so, because river courses and groundwater reservoirs can and do change radically through time. This happens both as a result of internal factors (for example, rivers migrating) and as a response to external forces (for instance, to name just two examples on opposite sides of the duration spectrum, earthquakes and tectonic subsidence). Although I do not work on the subject, I imagine there are many geomorphological studies reconstructing previous courses of North American rivers that could be combined with the same census data used in the manuscript to greatly improve this study.

Thus, my recommendation is REJECT, but I believe a substantial revision which takes into account the histories/movements of the different rivers (and, if at all possible, although I realise this will be harder, groundwater reservoirs) would warrant a RESUBMISSION.

Reviewer #3 (Remarks to the Author):

This is an important paper investigating the history of human settlements near water in the US from 1790 to 2010. It is important because it provides evidence for assertions many make and think are self-evident, and investigates regional differences in patterns. I have a few minor items and two moderate issues of concern.

Minor items:

1. There is an extra comma after the word "source" in the first paragraph after the abstract.
2. It would be helpful to provide a brief explanation of why the 1960 census data are not usable/available.
3. I think a better color ramp is needed for figure 1A. It is difficult to discern between the 3 blue classes and the two green ones.
4. The text for figure 3 should explain what the data points are (DMR classes, I'm assuming, for 3A? The census year for 3C?) And what the colors mean. Is red to blue simply the decades between 1790 and 1870? Why does 3A. progress red to blue but then 3B go blue to red?
5. Similarly the text for figure 4 should explain what the data points are. Also why does 4F only have data since ~1830? (I'm assuming, as a western frontier, that there were no settlements there prior to that date).
6. Again, text for supplementary figures should indicate what the data points are (e.g., I'm assuming S1B are DMR classes again?).

Moderate issues:

1. Is there a relationship between aquifer type and river density? I'm wondering if correlation between these two could confound analysis. A brief treatment in the text would probably suffice.
2. I disagree with the evaluation that "major aquifers with high recharge rates, especially those at recharge rates of 100-300 mm/a, were associated with low desirability of living close to major rivers (Supplementary Fig. 2)." This figure seems to show that *only* the 100-300 mm/a aquifer type shows a low desirability (positive slope), while all the others should a generally negative slope.

I believe all of these issues are fixable and do not detract from the overall importance of this paper.

Elisabeth Larson

libby.larson@nasa.gov

Reviewer #4 (Remarks to the Author):

This study addresses the evolution of human population distance to water for the conterminous US over the period of 1790-2010. This is an innovative and important research question in the newly developed discipline of socio-hydrology. The authors have made excellent headway in linking humans and water and provided a quantitative demonstration that human population preferred to live close to major rivers in pre-industrial periods and have moved to areas overlying major aquifers since industrial revolution. This paper makes an exciting contribution to the study of Socio-hydrology by incorporating spatial perspective and capturing long-term (over two centuries) human-water coevolution based on available historical data. The method is well described and based on solid scientific background. The topic of the paper can appeal a wide range of readers who are working on water resources sustainability, and broadly human-nature interaction. The subject matter is therefore suitable for Nature Communications.

Overall, the study is well designed and well written. With appropriate method and innovative results, this is an excellent paper and has very good potential for publication in Nature Communications. However, there are some parts which requires further discussion or elaboration. I recommend publication subject to the minor comments suggested below.

1)

This study showed the evolution of human water distance for the conterminous US influenced by societal progress and technological development, including industrial revolution and groundwater pumping technique. However, the uncertainty could come from water diversion projects especially during the recent several decades (maybe half

century). I acknowledge that the authors explained the importance of water diversion. For example, in Line 163, “water supply in post-industrial societies has evolved from “people go to water” to “water goes to people”, and it is understandable that the current knowledge and data availability might limit a further quantitative analysis for water diversion issue over the whole conterminous US. However, it would be helpful to include some water diversion projects and discuss their impact on human water distance here.

2)

This paper analyzed the temporal trends of human population distribution in relation to both major rivers and groundwater for the conterminous US. The dynamic trend of the importance of major rivers was nicely captured by the relationship between normalized population density and distance to major rivers as well as their slopes (Figure 3, a-c).

The finding with regard to aquifer type is interesting and useful- the advanced groundwater pumping technology around 1940 influenced the human population distribution overlying aquifers. However, the statement on groundwater is somewhat weak, since the turning point of the year of 1940 was currently identified through eyeballing (Figure 3, d). I would like to see a quantitative basis of this to strengthen the statement.

3)

The paper presented 3 major typologies of temporal changes in NPD vs DMR and related them to the local settlement history, water resource endowments, and climate condition. This revealed regional heterogeneity and enriched the analysis. However, it is not clear what this result can be used for. I would therefore suggest that the authors add some discussion on the broader impacts or relevance with respect to the result on regional heterogeneity.

4)

Line 197: "Ensuring adequate infrastructure investment is therefore extraordinarily essential to sustain stable water supplies and decrease the vulnerability of water supply systems." Should the vulnerability of water supply systems get increased with increasing investment?

Reviewer #2 (Remarks to the Author):

This article tackles an interesting question, that of the evolving relationship between human populations and water, by combining information on past censuses and on freshwater source location. The methods are well explained and the paper is easy to read.

Regrettably, the authors "assumed temporal stationarity of the spatial distribution of both major rivers and groundwater over time" (manuscript lines 265-266). This is simply not something that happens, which invalidates all their subsequent work: the quantitative aspects are not believable, nor is an estimate of the error given.

The authors should at least reference some geomorphological studies to support their course of action. However, I believe it will be hard to do so, because river courses and groundwater reservoirs can and do change radically through time. This happens both as a result of internal factors (for example, rivers migrating) and as a response to external forces (for instance, to name just two examples on opposite sides of the duration spectrum, earthquakes and tectonic subsidence). Although I do not work on the subject, I imagine there are many geomorphological studies reconstructing previous courses of North American rivers that could be combined with the same census data used in the manuscript to greatly improve this study.

Thus, my recommendation is REJECT, but I believe a substantial revision which takes into account the histories/movements of the different rivers (and, if at all possible, although I realise this will be harder, groundwater reservoirs) would warrant a RESUBMISSION.

RESPONSE: We appreciate the reviewer's suggestion to explicitly consider river course changes. We have incorporated a discussion (lines 211-213, 216-223) with appropriate findings from the river geomorphology literature and including data from global databases of river widths and reservoir construction. Specifically,

- 1) River course migration: We have considered the drivers of river course migration and compared published migration rates, with the overall conclusion that migration widths are much smaller than the resolution of our analysis (1 km). Thus, at this spatial scale the assumption of stationarity of river networks is reasonable. Please see Lines 211-213 and the Supplementary Note for details.**
- 2) River widths: Despite the generally minimal effect from river migration, we agree with the reviewer about the importance of the coverage of river courses. We have now included river width data from the Global River Widths from Landsat (GRWL) database, with approximately 7500 km of rivers with width larger than 1 km. This resulted in a small increase in the areas (0.8%) with distance to major rivers (DMR) of zero, compared to our previous result assuming a constant river width of 1 km. (Lines 281-288)**
- 3) Reservoirs: Again, inspired by the reviewer's comments, we now include the temporal evolution of reservoirs from the Global Reservoir and Dam (GRanD) dataset to identify reservoirs based of the decade of construction. We use the reservoir areas from GRanD and again this resulted in a small increase (2.6%) in**

the areas with DMR of zero, compared to the reference river width based on GRWL data. Thus our calculated DMR for each decade now includes geomorphological alteration over time. (Lines 292-295)

- 4) Aquifers: The timescales for large-scale changes in aquifer boundary migration are geological rather than human in scale. However, the importance of aquifer volume changes due to groundwater pumping has been growing in recent decades, and is expected to continue to grow in significance in coming decades. We have added a discussion in lines 216-223.

After including river width and considering the growth of reservoirs, we re-calculated all of our results and made the associated changes in the text, Figures, and tables. However, the general results were still the same and our conclusions on human distance to water remained. See lines 105-111, 147, 154, 161, and 165-166 for the detailed modifications.

Reviewer #3 (Remarks to the Author):

This is an important paper investigating the history of human settlements near water in the US from 1790 to 2010. It is important because it provides evidence for assertions many make and think are self-evident, and investigates regional differences in patterns. I have a few minor items and two moderate issues of concern.

Minor items:

1. There is an extra comma after the word "source" in the first paragraph after the abstract.

RESPONSE: We appreciate very much that the reviewer scrutinized this and have removed the extra comma in Line 37.

2. It would be helpful to provide a brief explanation of why the 1960 census data are not usable/available.

RESPONSE: We agree with the reviewer and have added a brief explanation (lines 248-251): "Note our analysis excluded 1960 because these data are still not available due to the unusually restrictive data suppression strategy employed by the 1960 Census (https://www.nhgis.org/user-resources/faq#1960_Data)."

FYI, clipped from the link above:

What is wrong with the 1960 data? Back to Top

Sadly, such a common question! The 1960 Census employed a uniquely restrictive data suppression strategy that leaves many data tables with lots of missing data. In addition, the NHGIS can offer only a small set of data tables for states and counties due to a scarcity of digital 1960 Census data, and the 1960 tract data come from 2 separate sources, resulting in some inconsistent redundancy. Review the Tabular Data Documentation page for more detailed information.

The Minnesota Population Center is currently working in collaboration with the Census Bureau to recover lost data from the 1960 Census of Population and Housing. Once the project is completed, new 1960 summary files will be available along with additional microdata products.

3. I think a better color ramp is needed for figure 1A. It is difficult to discern between the 3 blue classes and the two green ones.

RESPONSE: We agree with the reviewer. Also, the distance classes were hard to discern at the continental scale. We therefore have reduced the number of the distance classes to make their distribution patterns more discernible. The updated color ramp and distance classes are shown in Figure 1A below.

4. The text for figure 3 should explain what the data points are (DMR classes, I'm assuming, for 3A? The census year for 3C?) And what the colors mean. Is red to blue simply the decades between 1790 and 1870? Why does 3A. progress red to blue but then 3B go blue to red?

RESPONSE: As suggested by the reviewer, we have added an explanation to the caption that the data points for Figure 3a and 3b are based on DMR classes while those for Figure 3c and 3d are based on census year. These are also labeled on the X axes.

Also, as suggested by the reviewer, we have added an explanation to the caption regarding the color gradient. In Figure 3a we use the red-to-blue gradient to indicate

the trend for the decades to 1870 and in Figure 3b the gradient from blue to red to refer to decades from 1870. This choice is to highlight with the darker blues stronger desirability to live close to major rivers (steeper slopes) and the darker red colors relatively weaker desirability (smaller slopes). Also, we changed the color ramp for Figure 3d and increased the symbol size to make them easier to observe.

Figure 3 | The dynamic human population distribution in relation to major rivers and groundwater in the conterminous US. The relationship between normalized population density (NPD) and distance to major rivers (DMR) from **a** 1790 to 1870 (red to blue), and **b** 1870 to 2010 (blue to red). **c** The changing desirability of living close to major rivers, reflected in the slope of NPD vs DMR. **d** Mean population density overlying each aquifer type from 1790 to 2010. We use the red-to-blue gradient to indicate the trend for the decades to 1870 in **a** and the gradient from blue to red for decades from 1870 in **b**. The data points for a-b are based on DMR classes while those for c-d are based on census year.

5. Similarly the text for figure 4 should explain what the data points are. Also why does 4F

only have data since ~1830? (I'm assuming, as a western frontier, that there were no settlements there prior to that date).

RESPONSE: As suggested by the reviewer we have added further explanation to the caption, similar to the above comment. we previously included slopes with $p < 0.05$. As the reviewer suggested, we have systematically determined the starting years of analysis for each HUC based on their settlement history and only included data in Figure 4d-f after these initial years. This is now explained in the methods (line 327-331) and in the caption of Figure 4. We have added the initial decade of analysis by HUC in new supplementary Figure 7 and their dynamic settlement area percentages in supplementary Table 4.

Lines 327-331: "Only statistically significant slopes ($p < 0.05$) between NPD and DMR were used. Due to the westward migration of primarily European settlers, the settled portions of each HUC expanded gradually over time. The initial decade of analysis was determined for each HUC based on the decade when the settled area reached 90% of the total area (Supplementary Table 4 and Supplementary Fig. 7)."

Figure 4 | Three major types of trajectories of human water coevolution, based on the relationship between normalized population density (NPD) and distance to major rivers (DMR), with HUC 1, 3, and 15 as examples. a HUC 1, 1850 to 2010, b HUC 3, 1850 to 2010,

and c HUC 15, 1870 to 2010. d-f The dynamics of the slope of NPD vs DMR for HUC 1, 3, and 15, 1790-2010. Only slopes with $p < 0.05$ during the settlement periods are shown. The data points for a-c are based on DMR classes while those for d-f are based on census year.

Supplementary Table 4. The settlement area percentage for each HUC in the conterminous US, from 1790 to 2010.

Decade	HUC1	HUC2	HUC3	HUC4	HUC5	HUC6	HUC7	HUC8	HUC9	HUC10	HUC11	HUC12	HUC13	HUC14	HUC15	HUC16	HUC17	HUC18
1790	99.9	99.9	38.5	20.8	48.9	13.1	0.0	1.4	0.0	0.0	0.0	0.0	0.0	0.0	0.0	0.0	0.0	0.0
1800	100.0	99.9	39.1	100.0	97.7	34.9	49.7	5.4	13.5	0.0	0.0	0.0	0.0	0.0	0.0	0.0	0.0	0.0
1810	100.0	100.0	44.3	78.7	90.0	59.2	53.7	47.4	13.5	4.0	14.1	0.1	0.0	0.0	0.0	0.0	0.0	0.0
1820	100.0	99.9	63.6	90.3	97.3	82.7	51.8	71.8	13.5	6.7	17.6	1.3	0.0	0.0	0.0	0.0	0.0	0.0
1830	100.0	100.0	87.2	100.0	99.2	89.0	57.0	85.3	13.5	7.2	20.9	1.3	0.0	0.0	0.0	0.0	0.0	0.0
1840	100.0	100.0	100.0	92.3	100.0	100.0	81.9	99.9	33.4	8.0	19.3	1.3	0.0	0.0	0.0	0.0	0.0	0.0
1850	100.0	100.0	100.0	100.0	100.0	100.0	92.0	100.0	98.5	20.2	38.2	96.7	58.4	7.3	2.0	11.6	97.5	96.8
1860	100.0	99.9	100.0	96.6	100.0	100.0	98.8	99.9	46.6	28.2	44.4	73.6	94.3	96.3	100.0	96.2	99.8	100.0
1870	100.0	100.0	100.0	97.1	100.0	100.0	100.0	99.9	89.7	96.8	49.6	84.6	100.0	98.8	94.0	98.8	100.0	100.0
1880	100.0	99.9	100.0	100.0	100.0	100.0	100.0	99.9	80.6	93.2	69.0	94.8	99.8	100.0	100.0	100.0	100.0	100.0
1890	100.0	99.9	100.0	100.0	100.0	100.0	100.0	99.9	99.2	98.1	75.6	98.5	100.0	100.0	100.0	100.0	99.7	100.0
1900	100.0	99.9	100.0	100.0	100.0	100.0	100.0	99.9	100.0	100.0	99.7	100.0	100.0	100.0	100.0	100.0	100.0	100.0
1910	100.0	99.9	100.0	100.0	100.0	100.0	100.0	99.9	100.0	100.0	100.0	100.0	100.0	100.0	100.0	100.0	100.0	100.0
1920	100.0	99.9	100.0	100.0	100.0	100.0	100.0	99.9	100.0	100.0	100.0	100.0	100.0	100.0	100.0	100.0	100.0	100.0
1930	100.0	99.9	100.0	100.0	100.0	100.0	100.0	99.9	100.0	100.0	100.0	100.0	100.0	100.0	100.0	100.0	100.0	100.0
1940	100.0	99.9	100.0	100.0	100.0	100.0	100.0	99.9	100.0	100.0	100.0	100.0	100.0	100.0	100.0	100.0	100.0	100.0
1950	100.0	99.9	100.0	100.0	100.0	100.0	100.0	99.9	100.0	100.0	100.0	100.0	100.0	100.0	100.0	100.0	100.0	100.0
1970	100.0	99.9	100.0	100.0	100.0	100.0	100.0	100.0	100.0	100.0	100.0	100.0	100.0	100.0	100.0	99.9	100.0	100.0
1980	100.0	99.9	100.0	100.0	100.0	100.0	100.0	100.0	100.0	99.9	100.0	100.0	99.9	100.0	100.0	99.9	99.9	100.0
1990	100.0	99.8	100.0	100.0	100.0	100.0	100.0	100.0	100.0	100.0	100.0	100.0	100.0	100.0	100.0	100.0	100.0	100.0
2000	100.0	99.9	100.0	100.0	100.0	100.0	100.0	100.0	100.0	100.0	100.0	100.0	100.0	100.0	100.0	100.0	100.0	100.0
2010	100.0	99.9	100.0	100.0	100.0	100.0	100.0	100.0	100.0	100.0	100.0	100.0	100.0	100.0	100.0	100.0	100.0	100.0

Note. The starting year of analysis was determined for each HUC as the year when the area percentage reached 90%.

Supplementary Figure 7. The initial decade of analysis for each HUC.

6. Again, text for supplementary figures should indicate what the data points are (e.g., I'm assuming S1B are DMR classes again?).

RESPONSE: As suggested by the reviewer, and similarly to the figures above, we have added explanatory text to the caption that these are the DMR classes.

Moderate issues:

1. Is there a relationship between aquifer type and river density? I'm wondering if correlation between these two could confound analysis. A brief treatment in the text would probably suffice.

RESPONSE: As suggested by the reviewer, we added additional explanatory text in the methods (lines 316-319) about this analysis that was previously shown in Supplementary Figure 2 and explained in lines 132-140. We combined DMR zones and aquifer types to analyze their interacting effect on population density.

"In addition, considering the complex interaction between rivers and aquifers, we calculated the NPD values associated with DMR for each of the 10 aquifer types and analyzed their combined effect on population density using 2010 as a snapshot"

2. I disagree with the evaluation that "major aquifers with high recharge rates, especially those at recharge rates of 100-300 mm/a, were associated with low desirability of living close to major rivers (Supplementary Fig. 2)." This figure seems to show that *only* the 100-300 mm/a aquifer type shows a low desirability (positive slope), while all the others should a generally negative slope.

RESPONSE: We agree with the reviewer, and we have now calculated the desirability of each aquifer type (indicated by the slope of NPD vs DMR) and listed the results in the new Supplementary Table 1. Also, we have modified the main text to clarify this point (Lines 133-140)

“Across the US, areas with the relatively low-yielding local and shallow aquifers (aquifer types 33 and 34), were associated with strong desirability of living close to rivers (slope ≤ -0.03 , Supplementary Fig. 2 and Table 1), regardless of recharge rate. Conversely, complex and major aquifers with high recharge rates (aquifer types 24, 13, 14, and 15) had relatively lower desirability of living close to rivers (slope ≥ -0.02). Major aquifers with recharge rates of 100-300 mm/a (aquifer type 14) had the lowest desirability of living close to major rivers, with NPD positively related with DMR, indicating the role of high-recharge major aquifers in facilitating the decoupling of the historical proximity of human settlements to rivers.”

Supplementary Table 1. Slope of NPD vs DMR in 2010 for each aquifer type

Aquifer types	Recharge rate (mm/a)	Area percentage (%)	slope	R ²	P value
Major groundwater basins					
11	<2	5.8	-0.047	0.95	0.00
12	2-20	8.4	-0.034	0.86	0.00
13	20-100	4.1	-0.015	0.20	0.04
14	100-300	10.9	0.019	0.60	0.00
15	≥ 300	0.4	-0.020	0.37	0.00
Aquifers with complex hydrogeological structures					
22	<20	14.1	-0.033	0.64	0.00
23	20-100	4.9	-0.043	0.91	0.00
24	100-300	14.6	-0.018	0.58	0.00
Local and shallow aquifers					
33	<100	20.3	-0.030	0.90	0.00
34	≥ 100	16.6	-0.033	0.95	0.00

I believe all of these issues are fixable and do not detract from the overall importance of this paper.

We appreciate the reviewer’s comment. We also thank the reviewer for mentioning the western frontier, which inspired us to consider the varied settlement history of HUCs and systematically modify the initial decade of analysis.

Reviewer #4 (Remarks to the Author):

This study addresses the evolution of human population distance to water for the conterminous US over the period of 1790-2010. This is an innovative and important research question in the newly developed discipline of socio-hydrology. The authors have made excellent headway in linking humans and water and provided a quantitative demonstration that human population preferred to live close to major rivers in pre-industrial periods and have moved to areas overlying major aquifers since industrial revolution. This paper makes an exciting contribution to the study of Socio-hydrology by incorporating spatial perspective and capturing long-term (over two centuries) human-water coevolution based on available historical data. The method is well described and based on solid scientific background. The topic of the paper can appeal a wide range of readers who are working on water resources sustainability, and broadly human-nature interaction. The subject matter is therefore suitable for Nature Communications.

Overall, the study is well designed and well written. With appropriate method and innovative results, this is an excellent paper and has very good potential for publication in Nature Communications. However, there are some parts which requires further discussion or elaboration. I recommend publication subject to the minor comments suggested below.

1) This study showed the evolution of human water distance for the conterminous US influenced by societal progress and technological development, including industrial revolution and groundwater pumping technique. However, the uncertainty could come from water diversion projects especially during the recent several decades (maybe half century). I acknowledge that the authors explained the importance of water diversion. For example, in Line 163, “water supply in post-industrial societies has evolved from “people go to water” to “water goes to people”, and it is understandable that the current knowledge and data availability might limit a further quantitative analysis for water diversion issue over the whole conterminous US. However, it would be helpful to include some water diversion projects and discuss their impact on human water distance here.

RESPONSE: We agree with the reviewer on the importance of water diversion projects and thus elaborated our discussion in Lines 183-192. Specifically, we used the city of Los Angeles as an example to illustrate the water resource development history and show the increasing distance between water sources and the cities. We also cited two studies that demonstrate the importance of infrastructure, including water diversion projects, for satisfying water demand for US cities. Such water diversion projects weaken constraints of local water resources in influencing human settlement locations.

“For example, in Los Angeles since 1900, as the population has grown, the water supply from the local Los Angeles River became limiting and the city began to import water from hundreds of kilometers away, including the Owens Valley in 1913, Mono Basin in 1940, the Colorado River in 1941, and the rivers of the Central Valley in 1971⁷. Modern cities can bring adequate water from previously remote and inaccessible sources by building canals or by groundwater pumping^{8,9,27,28}, increasing water availability and greatly reducing the portion of the total population considered at risk for water scarcity²⁶. Around 80% of large cities globally would have to travel less than 22 km to reach a potential unstressed water source adequate for several million people⁹. Thus, water transfers have contributed to the increase in the distance of cities from water sources.”

2) This paper analyzed the temporal trends of human population distribution in relation to both major rivers and groundwater for the conterminous US. The dynamic trend of the importance of major rivers was nicely captured by the relationship between normalized population density and distance to major rivers as well as their slopes (Figure 3, a-c). The finding with regard to aquifer type is interesting and useful—the advanced groundwater pumping technology around 1940 influenced the human population distribution overlying aquifers. However, the statement on groundwater is somewhat weak, since the turning point of the year of 1940 was currently identified through eyeballing (Figure 3, d). I would like to see a quantitative basis of this to strengthen the statement.

RESPONSE: We appreciate the reviewer for recognizing the importance of our results. As suggested by the reviewer, we have strengthened the quantitative analysis of the groundwater data. We greatly appreciate the reviewer’s suggestion because while the conclusion remains the same, our result is now much more robust.

First, we calculated the growth rate for each aquifer type based on fitted power functions. Major aquifers with recharge rate of more than 100 mm/a had the highest growth rates in mean population density.

Second, we used the Pettitt test to identify the turning point in growth rate based on of the dynamic differences in mean population density between the aquifer type with the highest growth rate (major aquifers with recharge rate larger than 300 mm/a, aquifer type 15) and the remaining nine types of aquifers. Considering the settlement history, the test was applied to decades after 1880. This result supported that the turning points were 1940 for all the nine aquifers ($P < 0.05$).

The related modification in the manuscript include:

In lines 308-313 of the methods: “We fit the time series of mean population density with a power function ($MPD = r_0 r^{t-t_0}$, for $t_0 \leq t \leq 2010$) and calculated the growth rate (r) for each aquifer type. The initial year, t_0 , was

based on the settlement history of each aquifer type. We used the Pettitt test to identify the turning point of the dynamic differences in mean population density between major aquifers with recharge rate greater than 300 mm/a (aquifer type 15) and the remaining nine types of aquifers.”

In lines 122-131 of the results:

“Higher population densities were found above aquifers with higher recharge rates (Fig. 3d). Major aquifers with high recharge rates (> 100 mm/a, aquifer types 14 and 15), were the most attractive for humans in the 20th century. The major aquifers with the highest recharge rate (aquifer type 15) supported the highest population densities and experienced the fastest growth in population density (Fig. 3d and Supplementary Table 2). The differences in population density between aquifer type 15 and the other nine types of aquifers increased steadily, but with a turning point identified in the 1940s ($p < 0.05$), when groundwater pumping became more efficient and advanced¹⁵. Generally, mean population density increased with potential water supply from underlying aquifers. With similar recharge rates, population density tended to be higher over major aquifers, and lower over local and shallow aquifers.”

Supplementary Table 2. Increasing mean population density from t_0 to 2010 for each aquifer type.

Aquifer types	Area						
Groundwater zone	Recharge rate (mm/a)	percentage (%)	r_0	r	t_0	R^2	p value
Major groundwater basins							
11	<2	5.8	0.101	1.036	1860	0.99	0.00
12	2-20	8.4	0.126	1.032	1850	0.94	0.00
13	20-100	4.1	0.642	1.027	1850	0.90	0.00
14	100-300	10.9	2.661	1.017	1800	0.99	0.00
15	≥300	0.4	0.698	1.043	1850	0.88	0.00
Aquifers with complex hydrogeological structures							
22	<20	14.1	0.237	1.025	1870	0.90	0.00
23	20-100	4.9	0.575	1.029	1820	0.81	0.00
24	100-300	14.6	2.337	1.018	1800	0.91	0.00
Local and shallow aquifers							
33	<100	20.3	0.472	1.030	1850	0.87	0.00
34	≥100	16.6	3.930	1.013	1800	0.98	0.00

3) The paper presented 3 major typologies of temporal changes in NPD vs DMR and related them to the local settlement history, water resource endowments, and climate condition. This revealed regional heterogeneity and enriched the analysis. However, it is not clear what this result can be used for. I would therefore suggest

that the authors add some discussion on the broader impacts or relevance with respect to the result on regional heterogeneity.

RESPONSE: We are deeply grateful to the reviewer for this comment. We found these regional differences to be very important and upon re-reading our manuscript we are honestly surprised at ourselves for not explaining this sufficiently previously. Thus, as suggested by the reviewer, we have added the following additional points to support the implications of the regional heterogeneity results.

The first type (Lines 148-151): “This relative stability may be indicative of an equilibrium state that follows the evolution from frontier to cities illustrated in Figure 2a. For example, by 1850 the New England settlements of HUC 1 had already been long-established with relatively little structural change since then.”

The second type (Lines 154-158): “This indicates decreasing desirability of proximity to rivers, with humans moving farther from rivers, which could be associated with relatively later settlement histories and thus continued evolution from towns to cities (Figure 2a) during the study period. However, this pattern could also be influenced by the presence of abundant groundwater resources (Figure 2c) as in HUC 3.”

The third type (Lines 161-164): “This pattern is mostly associated with arid regions and highlights that under conditions of water scarcity, the general hypotheses described in Figure 2 appear to be outweighed by the continued attractiveness of proximity to rivers.”

Discussion (Lines 199-203): “Regions (here based on HUCs) such as New England with long settlement history and early industrialization had stable distances from rivers with early evolution from frontier to cities (Figure 2a). Regions with temporally increasing desirability of proximity to rivers were located in the arid west, while regions with decreasing desirability were mostly found in humid regions with less vulnerability to water scarcity.”

4) Line 197: “Ensuring adequate infrastructure investment is therefore extraordinarily essential to sustain stable water supplies and decrease the vulnerability of water supply systems.” Should the vulnerability of water supply systems get increased with increasing investment?

RESPONSE: As implied by the reviewer, we have modified the sentence to make our point more clear by removing the final clause which may have been redundant (Line 235).

“Ensuring adequate infrastructure investment is therefore extraordinarily essential to sustain stable water supplies.”

Reviewer #2 (Remarks to the Author):

As I said in my previous review, I do not work on North American rivers, and thus had no clear idea of their migration speeds. The authors have worked diligently to address my doubts, and I find their article stronger for it.

However, I still believe there might be some distortion in their data due to earthquakes, since these can shift river courses dramatically (the 1906 San Francisco earthquake comes immediately to mind), so perhaps a short note recognising this somewhere in the article is in order.

In any case, the article seems fine enough to me (although I confess I have only re-reviewed the parts indicated by the authors as concerning my previous review), and I recommend ACCEPTING it.

Reviewer #3 (Remarks to the Author):

Thank you for addressing my concerns raised in the first review. I feel the paper has been strengthened substantially both by addressing my concerns and those of the other reviewers.

My recommendation is to ACCEPT for publication.

Elisabeth K. Larson

Reviewer #4 (Remarks to the Author):

The revision of the manuscript “The Evolution of Human Population Distance to Water in the USA from 1790 to 2010” has thoroughly addressed my original suggestions and comments. The statement on groundwater is now strengthened with quantitative support and the discussions around water diversions and regional heterogeneity are much improved.

It is also good to see the authors have extensively re-conducted the analyses by including river width and the dynamic coverage of river courses influenced by reservoirs over time. Together with the compiled migration rates from literatures, this can effectively support the assumption of “stationarity”, especially considering the resolution of this study.

Overall, the authors have done a wonderful job in revising and improving the manuscript. I suggest accepting the manuscript for publication in Nature Communications.

REVIEWERS' COMMENTS:

Reviewer #2 (Remarks to the Author):

As I said in my previous review, I do not work on North American rivers, and thus had no clear idea of their migration speeds. The authors have worked diligently to address my doubts, and I find their article stronger for it.

However, I still believe there might be some distortion in their data due to earthquakes, since these can shift river courses dramatically (the 1906 San Francisco earthquake comes immediately to mind), so perhaps a short note recognising this somewhere in the article is in order.

RESPONSE: We understand the reviewer's concern and have added text and references on this point in the Supporting Information. Very similar to our response for overall migration of rivers from natural or anthropogenic causes, these effects are very localized and for the entire spatial and temporal scope of our study, these effects are undetectable. We have added three additional supporting references as well as the following statement.

“Also note that seismic activity may contribute to landslide-induced flooding⁵, changes to river bed morphology⁶, and also rare documented changes to river courses⁷. These effects may become important in localized studies of seismically active zones, however at continental scales the integral effect is expected to be negligible.”

5. Dai, F. C., Lee, C. F., Deng, J. H., & Tham, L. G. The 1786 earthquake-triggered landslide dam and subsequent dam-break flood on the Dadu River, southwestern China. *Geomorphology*, 65, 205-221 (2005).
6. Field, M. E., Gardner J. V., Jennings A. E., & Edwards B. D.. Earthquake-induced sediment failures on a 0.25° slope, Klamath River delta, California. *Geology* 10, 542-546 (1982).
7. Sirovich, L., & F. Pettenati. Source inversion of the 1570 Ferrara earthquake and definitive diversion of the Po River (Italy). *Journal of Geophysical Research: Solid Earth* 120, 5747-5763 (2015).

In any case, the article seems fine enough to me (although I confess I have only re-reviewed the parts indicated by the authors as concerning my previous review), and I recommend ACCEPTING it.

RESPONSE: We thank the reviewer for finding our revised manuscript suitable for publication in Nature Communications now and appreciate the good suggestions.

Reviewer #3 (Remarks to the Author):

Thank you for addressing my concerns raised in the first review. I feel the paper has been strengthened substantially both by addressing my concerns and those of the other reviewers.

My recommendation is to ACCEPT for publication.

Elisabeth K. Larson

RESPONSE: Thank you very much for your positive comment on this study.

Reviewer #4 (Remarks to the Author):

The revision of the manuscript "The Evolution of Human Population Distance to Water in the USA from 1790 to 2010" has thoroughly addressed my original suggestions and comments. The statement on groundwater is now strengthened with quantitative support and the discussions around water diversions and regional heterogeneity are much improved.

It is also good to see the authors have extensively re-conducted the analyses by including river width and the dynamic coverage of river courses influenced by reservoirs over time. Together with the compiled migration rates from literatures, this can effectively support the assumption of "stationarity", especially considering the resolution of this study.

Overall, the authors have done a wonderful job in revising and improving the manuscript. I suggest accepting the manuscript for publication in Nature Communications.

RESPONSE: We thank Reviewer #4 for his/her appreciation of our study.